# Stereotactic MR-Guided Radiotherapy for Liver Metastases: First Results of the Montpellier Prospective Registry Study

**DOI:** 10.3390/jcm12031183

**Published:** 2023-02-02

**Authors:** Karl Bordeau, Morgan Michalet, Aïcha Keskes, Simon Valdenaire, Pierre Debuire, Marie Cantaloube, Morgane Cabaillé, William Jacot, Roxana Draghici, Sylvain Demontoy, Xavier Quantin, Marc Ychou, Eric Assenat, Thibault Mazard, Ludovic Gauthier, Marie Dupuy, Boris Guiu, Céline Bourgier, Norbert Aillères, Pascal Fenoglietto, David Azria, Olivier Riou

**Affiliations:** 1Montpellier Cancer Institute (ICM), University Federation of Radiation Oncology of Mediterranean Occitanie, University Montpellier, INSERM U1194 IRCM, 34298 Montpellier, France; 2Medical Oncology Department, Montpellier Cancer Institute (ICM), University Montpellier, 34298 Montpellier, France; 3Medical Oncology Department, CHU St. Eloi, 34000 Montpellier, France; 4Biometrics Unit, Montpellier Cancer Institute (ICM), University Montpellier, 34298 Montpellier, France; 5Radiology Department, CHU St. Eloi, 34000 Montpellier, France

**Keywords:** magnetic resonance-guided radiotherapy (MRgRT), stereotactic body radiotherapy (SBRT), liver metastases, liver tumors

## Abstract

Liver stereotactic body radiotherapy (SBRT) is a local treatment that provides good local control and low toxicity. We present the first clinical results from our prospective registry of stereotactic MR-guided radiotherapy (MRgRT) for liver metastases. All patients treated for liver metastases were included in this prospective registry study. Stereotactic MRgRT indication was confirmed by multidisciplinary specialized tumor boards. The primary endpoints were acute and late toxicities. The secondary endpoints were survival outcomes (local control, overall survival (OS), disease-free survival, intrahepatic relapse-free survival). Twenty-six consecutive patients were treated for thirty-one liver metastases between October 2019 and April 2022. The median prescribed dose was 50 Gy (40–60) in 5 fractions. No severe acute MRgRT-related toxicity was noted. Acute and late gastrointestinal and liver toxicities were low and mostly unrelated to MRgRT. Only 5 lesions (16.1%) required daily adaptation because of the proximity of organs at risk (OAR). With a median follow-up time of 17.3 months since MRgRT completion, the median OS, 1-year OS and 2-year OS rates were 21.7 months, 83.1% (95% CI: 55.3–94.4%) and 41.6% (95% CI: 13.5–68.1%), respectively, from MRgRT completion. The local control at 6 months, 1 year and 2 years was 90.9% (95% CI: 68.3–97.7%). To our knowledge, we report the largest series of stereotactic MRgRT for liver metastases. The treatment was well-tolerated and achieved a high LC rate. Distant relapse remains a challenge in this population.

## 1. Introduction

Although the metastatic course of a cancer affects overall prognosis and survival rates, local treatment of metastases is an issue because it may allow for better control of the disease in an oligo-metastatic stage [1]. The most commonly individualized tumors in the liver are secondary tumors, mainly of colorectal, pulmonary and mammary origin. Their oncological prognosis is dependent on the primary site, but remains poor. Surgery is currently the cornerstone of curative treatment of these liver tumors [2,3,4]. In the case of unresectable or inoperable patients due to predictable post-surgical liver failure, macrovascular invasion, comorbidities or impaired general condition, focal ablative therapies are the preferred options. Among these focal ablative techniques, stereotactic body radiotherapy (SBRT) is a technique of choice because it provides good local control and low toxicity. Even though liver SBRT is a more recently applied technique in this location, it has many advantages that allow it to create a place in the available therapeutic arsenal. Liver SBRT is a noninvasive technique that can be applied on an outpatient basis and is easy to combine sequentially with systemic treatments because of its excellent tolerance. The indication of SBRT for liver metastasis in the current international guidelines is not consensual. Liver metastasis SBRT should be performed in high-volume radiotherapy centers because the procedure is usually complex. This is indeed one of the most difficult locations for SBRT because hepatic respiratory movements must be taken into account in the planning and delivery of the treatment. Despite interesting prospective data and validation by some international guidelines that recognize it as an effective and safe technique, there is a heterogeneity of technologies and a lack of prospective evaluation of liver SBRT [5,6]. In recent years, magnetic resonance-guided radiotherapy (MRgRT) has opened up new possibilities for further technological improvements in liver SBRT [7]. First, upper abdominal malignant lesions, like liver metastases, suffer from poor tissue contrast in X-ray imaging. The anatomy of the liver and radiosensitive structures like the duodenum and stomach are reasonably refined by MRI contrast, even on 0.35T MR-linear accelerators, reducing treatment uncertainties and margins. Second, breathing and diaphragm movement frequently induce large intrafraction translation [8]. With online cine MR sequences, MR Linac systems combine gating (breath hold treatments) and a live tracking technique (comparison of the deformed contours of the tracked target with its initial static anatomy). The beam is turned off when the tracked structure is outside the margin predetermined by its initial position. Third, stereotactic magnetic resonance adaptive radiotherapy (SMART) is useful for upper abdominal malignancies located around radiosensitive organs at risk (OAR), especially pancreatic tumors [9,10]. This procedure is not required for all liver tumors, but is for lesions located very close to organs at risk that can be spared, such as the heart, duodenum, stomach, and small and large intestine [11]. Few MRgRT retrospective studies have published promising results for liver metastatic lesions. Local control is excellent and toxicities are low, but study power has been limited [12,13]. Thus, it is a new and innovative technique, but published data are still scarce.

The purpose of this study is to present toxicities and oncological outcomes in our first prospective registry study for the treatment of liver metastases by MRgRT, in order to evaluate by complementary data the value of this technique. 

## 2. Methods and Materials

### 2.1. Patient Selection

All patients treated for liver metastases from October 2019 to April 2022 were included in this prospective registry study. Stereotactic MRgRT indication was confirmed by multidisciplinary specialized tumor boards, and a secondary technical board (radiation oncologists and physicists) checked stereotactic MRgRT eligibility. The inclusion criteria were synchronous or metachronous oligo-metastatic liver metastases and oligo-progressive liver metastases, all from various primary cancers. The contraindications comprised ECOG (Eastern Cooperative Oncology Group) > 2, non-MRI-compatible pacemaker, age < 18 years, unstable psychiatric disease, and other MRI contraindications.

This study was registered in the Health Data Hub (registration number: #1802) and was approved by our local research committee (2020/01). All patients signed an informed consent form before treatment. 

### 2.2. Radiotherapy Planning and Delivery 

Our general treatment planning, breath hold procedure and daily adaptive workflow (when necessary) for MRgRT have been already published [9,10,14]. However, in order to facilitate the reading of this article, we briefly recall here the course and the characteristics more specific to the hepatic location. 


Image acquisitions for planning and treatment


Patients underwent contrast-enhanced CT simulation directly followed by 0.35T MRI simulation using the MRIdian^®^. Registration with contrast-enhanced 1.5T MRI in planning conditions was mandatory in order to optimize tumor location. The 0.35T MRI images were made with true fast imaging with steady-state free precession (TRUFISP) sequences (T1/T2 weighted). Acquisitions were performed using a breath hold technique, mainly physiologic end-expiration, 17 to 25 s, 1.6 × 1.6 × 3 mm^3^ or 1.5 × 1.5 × 3 mm^3^ resolution, 45 × 45 × 24 to 54 × 47 × 43 cm^3^ maximum fields of view. Radiotherapists carried out respiratory coaching and continuous verification of the reproducibility of the positioning and breathing thanks to continuous cine MR acquisitions. A 0.35 T MRI breath hold acquisition identical to the planning acquisition was performed for each treatment fraction. The radiotherapists performed a rigid registration of the tumor volume and the liver with medical validation by the radiation oncologist. 


Target volume and organs at risk (OAR) contouring


Liver metastases (GTV) were delineated in CT, MRIdian^®^ simulation images and planning 1.5T MRI images. Any other useful diagnosis imaging techniques were used. A planning target volume (PTV) was created by adding a 5 mm isotropic extension from the GTV. The following OAR were delineated on all available slices: liver, spinal cord, esophagus, right and left kidneys, stomach, duodenum, small and large intestine, heart, and right and left lungs.


Dosimetry


Additional volume structures were generated to facilitate the planning procedure. Among these, the (liver—PTV) and (liver—GTV) volumes were used. Other structures could be used depending on the needs and the location of the lesion. Dosimetry was performed on the Viewray^®^ treatment planning system (TPS) with a Monte Carlo algorithm. Our target volume objective was to achieve 95% PTV coverage within the 95% isodose. Treatment was done with step-and-shoot IMRT with 6 MV photons. 


Optional adaptive procedure


In case of OAR proximity to PTV, an optimized PTV (PTV opt) was created by excluding OAR plus 5 mm from the PTV, and patients were treated with daily adaptive fractions. Rigid registration of the GTV and OAR contours propagation on the MR image of the day were performed with deformable image registration. OAR contours were modified to the daily anatomy. Adaptation of the initial plans was done to get the best PTV coverage and OAR sparing. The electron density map (applied from the CT to the MR images) and the skin contour were modified to ensure correct dose calculation [15]. 


Continuous image guidance by Cine MR acquisitions


The GTV (if visible) or the whole liver on a sagittal slice containing the GTV were tracked on sagittal slices by cine MR. The following parameter was usually used for tracking: beam stopped when 5% or more of the structure was outside a 3 mm threshold from its initial position.

### 2.3. Clinical Assessment, Dosimetric Evaluation and Endpoints


Study objectives


The primary objective was to assess acute and late toxicities. The secondary objective was to assess survival outcomes (overall survival (OS), progression-free survival (PFS), local control (LC), and intrahepatic relapse-free survival (IHRFS)). 


Toxicity, survival and response evaluations


Common Terminology Criteria for Adverse Events (CTCAE) v5.0 were used to evaluate toxicities. Treatment response was evaluated using Response Evaluation Criteria in Solid Tumors v1.1. Local control was defined as the absence of progression within RECIST of the liver metastasis. Overall survival was defined as death (any cause) since the end of MRgRT. Progression-free survival was defined as relapse or death (any cause) since the end of MRgRT. Intrahepatic relapse-free survival was defined as a new metastasis inside the liver (outside the PTV). Clinical examination and radiological (CT, MRI or PET/CT) and biological (blood sample) assessment were performed every 3 months. Follow-up was done starting on the first day of MRgRT treatment and continued until death or the latest news for each patient. 

### 2.4. Statistical Analysis 

The number of observations (n) and their frequency (%) were used to describe qualitative variables. A median and range were recorded for quantitative variables from each patient’s baseline characteristics. An average and standard deviation were registered for dosimetric measures. 

Median follow-up and clinical outcomes (LC, OS, PFS, IHRFS) were estimated using the Kaplan–Meier method. 

Statistical analyses were performed using Stata v16.0 and GraphPad PRISM v9.4.

## 3. Results

### 3.1. Patient and Treatment Characteristics 

Between October 2019 and April 2022, 26 patients were treated for 31 lesions with stereotactic MRgRT. The median age was 68.5 years with a balanced sex ratio. The primary sites were varied, mostly colorectal (42.5%) and pancreatic (23.1%). Nearly 90% of the patients were already pretreated for one or more hepatic metastases, mainly by systemic treatment. The lesions were mainly located in the right liver (48.3%). The number of lesions treated by stereotactic radiotherapy was unique in nearly 80% of cases. The median sum of the lesion diameters was 21 mm (6.0; 70.0) (Table 1). The delivered dose was 50 Gy in 5 fractions for 54.8% of the lesions and was increased to 60 Gy for 25% of the lesions. The median volume of PTVs was 35.6 cc (9.9; 343.2). An adaptive protocol was required for 16.1% of the lesions. Figure 1 shows a dosimetry for a liver metastasis next to the heart, requiring adaptive treatment. The median treatment time was 48 min (37; 73). Table 2 summarizes the median doses delivered to target volumes and organs at risk.

### 3.2. Toxicities

No MRgRT-related acute toxicities of CTCAE grade 3 or higher were noted. The main acute toxicities were gastrointestinal: nausea or vomiting, grade 1 to 2 (34.7%), abdominal pain, grade 1 (11.4%) and diarrhea, grade 1 (7.7%). Regarding hepatobiliary events, one patient experienced a grade 4 acute angiocholitis complicated by septic shock, but this was probably unrelated to radiotherapy as the radiation fields were well away from the bile ducts. This event might better have been related to a history of surgery close to the bile ducts. This complication evolved to late grade 4 biliary stenosis that led to several other hospitalizations, with an unfavorable evolution. A grade 3 acute angiocholitis due to a biliary prosthesis obstruction unrelated to MRgRT was noted in another patient followed for a pancreatic adenocarcinoma. 

Late toxicities were mainly related to metastatic disease progression or systemic treatments and were marked by grade 1 to 2 abdominal pain (36.4%), grade 1 to 2 diarrhea (18.2%) and grade 1 to 2 nausea or vomiting (13.7%). On the hepatobiliary level, a late grade 3 angiocholitis occurred at 17 months of follow-up (in a patient treated for colorectal adenocarcinoma) secondary to ischemic stenosis of the upper bile duct, for which the imputability of radiotherapy is possible, but probably multifactorial. He had been treated by hepatic surgery on two occasions, and a radiofrequency ablation of segment IV, all located opposite the area of bile duct dilatation. Another patient followed for a pancreatic adenocarcinoma experienced a grade 3 angiocholitis explained by a local progression of the pancreatic tumor. 

The detail of acute and late clinical events can be found in Table 3.

### 3.3. Survival Analysis 

After a median follow-up of 17.3 months (95% CI: 6.1; 20.7), overall survival was 83.1% (95% CI: 55.3; 94.4) at 1 year and 41.6% (95% CI: 13.5; 68.1) at 2 years, for a median of 21.7 months (95% CI: 12.6; NE) from radiotherapy (Figure 2A). A total of 8 patients died during follow-up, six of whom died of progression of the underlying metastatic cancer (75%). Progression-free survival was 12.9% (95% CI: 3.3; 29.3) at 1 year and 8.6% (95% CI: 1.5; 23.9) at 2 years, for a median of 4.1 months (95% CI: 2.8; 5.8) (Figure 2B).

Local control was 90.9% (95% CI: 68.3; 97.7) at 6 months, 1 year and 2 years (Figure 2C). Three local recurrences occurred in the middle of the irradiation field, two of them in the same patient. The 2 patients were followed for pancreatic adenocarcinoma with hepatic metastasis and received 50 Gy in 5 fractions. Intrahepatic relapse-free survival was 61.9% (95% CI: 39.1; 78.3) at 6 months, 33.8% (95% CI: 14.4; 54.4) at 1 year and 25.3% (95% CI: 7.9; 47.6) at 2 years for a median of 9.4 months (95% CI: 4.8; 14.5) (Figure 2D). 

A univariate analysis was performed to look for prognostic factors for survival. No significant factors were found among age, performance status, type of primary, number of lesions, tumor volume or lesion diameter. A multivariate analysis was not performed due to small numbers.

## 4. Discussion

When looking at the clinical results of a new treatment technique, especially in radiotherapy, it is important to focus first on the tolerability of the treatment, to make sure that the technological developments studied go hand in hand with good tolerability of the newly performed technique. In this study, 26 patients were included, and 31 liver lesions were treated with MRgRT at a median dose of 50 Gy in five fractions. Severe grade 3+ liver toxicities were minimal. In total, one grade 3 late liver toxicity potentially attributable to radiotherapy (3.8%) was retained. Acute tolerance was excellent, in part due to monitoring by adaptive radiotherapy for 16% of patients. Rosenberg et al. published one of the first studies of stereotactic MR-guided radiotherapy on liver tumors, with a similar number of patients compared to our study. Hepatic tumors were mostly secondary (18 patients out of 26 treated, i.e., 69.2%). Severe toxicities were minimal. Two patients (7.7%) had experienced grade 3+ hepatobiliary toxicity. The treatments were not performed by adaptive radiotherapy [16]. Our study therefore represents, to date, despite its small size, the largest series of MRgRT for liver metastases. In 2022, a prospective, non-inferiority, randomized phase II trial has been initiated to include 82 patients with liver metastases [17]. The aim of this study is to compare stereotactic MR-guided adaptive radiotherapy with stereotactic radiotherapy based on a more conventionally mediated ITV strategy. An initial clinical experience and patient-reported outcomes were recently published on the first 20 patients, including 18 patients with liver metastases [18]. Another prospective phase II trial, named RASTAF-IRM, is currently including patients with liver tumors in France (ClinicalTrials.gov Identifier: NCT04242342). Final results are still pending, but these studies may validate the growing interest in hepatic adaptive radiotherapy. 

Furthermore, the most important result for studies of ablative treatment of metastases is the local control of the treated lesions, since it reports the local efficacy of this highly focused treatment, ignoring the evolution of the disease outside the treatment area. Liver SBRT is no exception to this rule, and local control is reported in most published studies. Local control in our study was 91% at 1 and 2 years. These results are consistent with the evidence in the literature. A meta-analysis of published data pooled from 18 prospective and retrospective studies from 2006 to 2017 (656 patients) of non-MR-guided stereotactic radiotherapy of metastases from colorectal primaries was recently published. Local control was 67% and 59.3% at 1 and 2 years, respectively, which is lower than the data of our retrospective study [19]. However, using a linear correlation model, the authors associated the biological equivalent dose with local control. Thus, a biological equivalent dose of 100 Gy on the tumor would improve local control and overall survival by 21% at 2 years, according to their calculations. This trend of improved local control, gained by increasing the biological equivalent dose with stereotactic radiotherapy, was also reported by another meta-analysis [20]. Local control at 3 years was improved by 28% in absolute value with a BED10 higher than 100 Gy. Local control of disease is all the more important as it seems to correlate with overall survival. Klement et al. analyzed oncologic data on more than 300 liver metastases of colorectal cancer and suggested an improvement in median overall survival in the absence of local recurrence (30.6 months vs. 25.4 months) [21]. In our study, we report three local failures of radiotherapy treatment, all in patients with liver metastases of pancreatic primaries. This confirms the interest in pursuing the customization of liver metastasis SBRT according to the origin of the primary. The good tolerance of the treatment allows us to consider dose escalation for the more radioresistant tumors, provided that they are in a favorable anatomical conformation at a distance from OAR. In this favorable clinical situation with PTVs more than 2 cm from critical organs, the RASTAF MRI trial mentioned above, in which we participate, proposes a dose of 60 Gy in 6 fractions in order to obtain a higher biological dose equivalent to reverse the radioresistance of certain tumors and improve local control.

Finally, although it is the largest series on the subject to date, we must acknowledge that our study has many limitations. In addition to the ambispective, monocentric and nonrandomized nature of the study, the number of patients included is small, and this makes these results difficult to generalize. Therefore, the results of the search for prognostic factors for overall survival were unfortunately non-contributory. Moreover, three local recurrences (Figure 3) of pancreatic primitives were identified in our series, all located in the radiation field, suggesting a lower tumor radiosensitivity in these patients. This opens a field of inquiry for dose customization according to the primary. For example, a radiosensitivity index has been proposed by genomic analysis, and it showed that histology was an important factor to consider in the delivered dose of SBRT [22,23].

## 5. Conclusions

We report the clinical results of the largest series to date of stereotactic MRgRT for the treatment of liver metastases. We confirm the feasibility and the good tolerance of this treatment in this indication. Acute and late gastrointestinal and liver toxicities were low and mostly unrelated to MRgRT. Only 5 lesions (16.1%) required daily adaptation because of the proximity of organs at risk (OAR). Local control is satisfactory (local control at 2 years of 90.9% (95% CI: 68.3–97.7%)), but metastatic relapse outside the treated volume remains a challenge in this population.

## Figures and Tables

**Figure 1 jcm-12-01183-f001:**
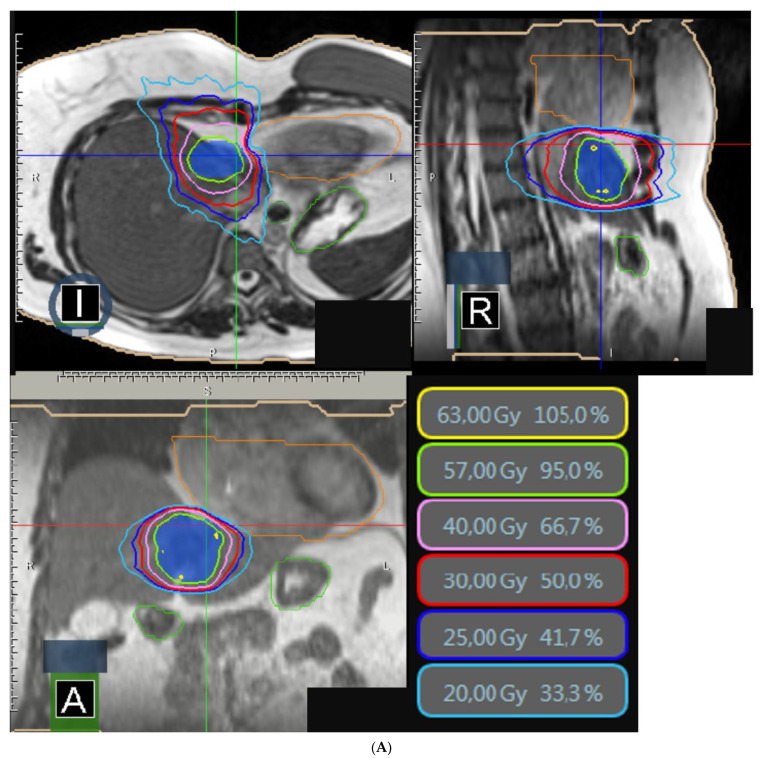
(**A**,**B**) Typical MRgRT dosimetry for liver metastases. (**A**): Example of dosimetry for a liver metastasis from breast cancer. We performed an adaptive process because of the proximity of the heart (in orange). PTV in blue colorwash. Isodose lines: 63 Gy in yellow, 57 Gy in green, 40 Gy in pink, 30 Gy in red, 25 Gy in blue, 20 Gy in cyan. Abbreviation: Gy = gray (**B**): Example of dosimetry for a liver metastasis from colorectal cancer. We performed an adaptive process because of the proximity of the heart (in pink) and stomach (in light blue). PTV in purple colorwash. Isodose lines: 47.5 Gy in green, 40 Gy in pink, 30 Gy in red, 25 Gy in blue, 20 Gy in cyan. Abbreviation: Gy = gray.

**Figure 2 jcm-12-01183-f002:**
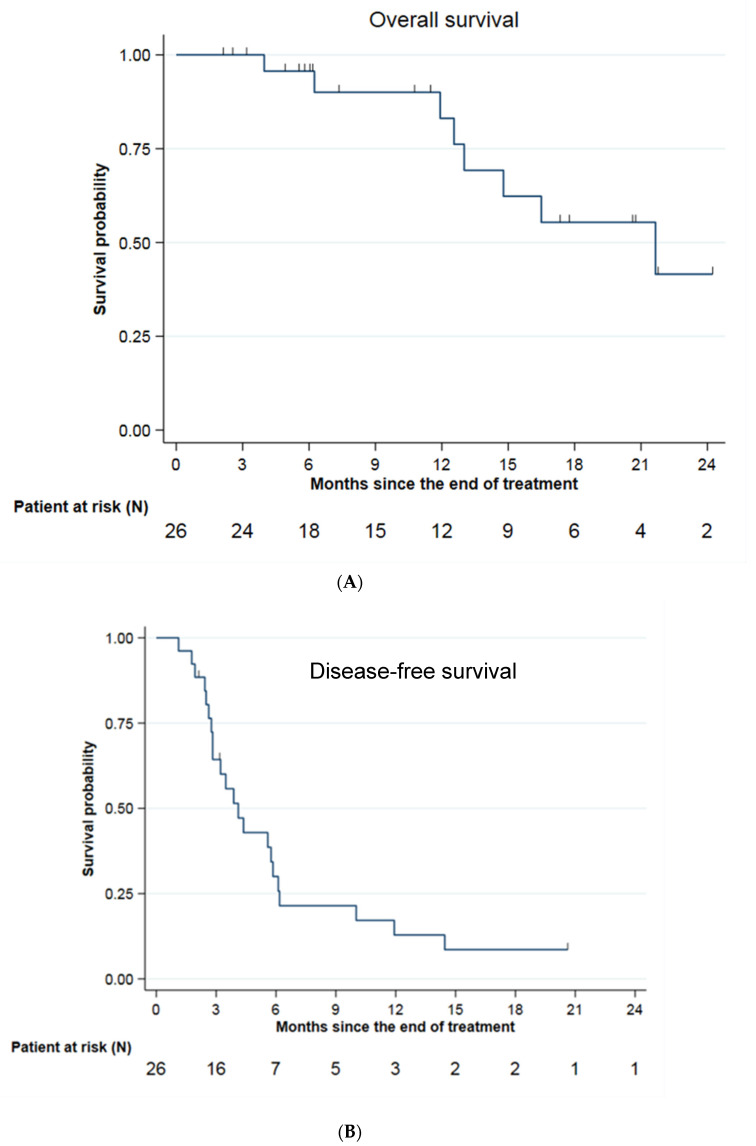
Survival data. (**A**): Overall survival; (**B**): Disease-free survival; (**C**): Local control; (**D**): Intrahepatic relapse-free survival.

**Figure 3 jcm-12-01183-f003:**
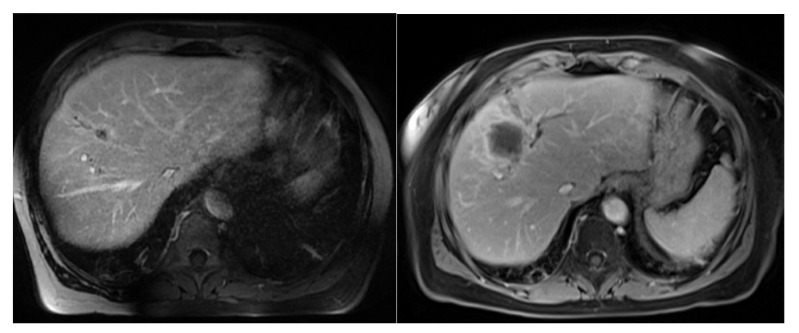
Local relapse: Example of a 15 mm MRI (**left** image) liver metastasis from pancreatic tumor treated by stereotactic MRgRT (50 Gy in 5 fractions). On the (**right** image), T1 portal gadolinium-enhanced MRI sequence shows a massive local recurrence 6 months after MRgRT.

**Table 1 jcm-12-01183-t001:** Patients’ baseline characteristics.

Sex	
*Women*	13 (50.0%)
*Men*	13 (50.0%)
Median age (range)	68.5 (45.0; 89.0)
Number of lesions	
*1* *2*	21 (80.8%) 5 (19.2%)
Primary cancer	
*NSCLC* *Colorectal* *Pancreas* *Ovary* *Sarcoma* *Breast* *Esophagus* *Kidney*	2 (7.7%) 11 (42.3%) 6 (23.1%) 1 (3.8%) 1 (3.8%) 3 (11.5%) 1 (3.8%) 1 (3.8%)
Number of previous liver local treatments	
*0* *1* *2*	3 (11.5%) 9 (34.6%) 14 (53.8%)
ECOG score	
*0* *1* *2*	12 (46.2%) 10 (38.5%) 4 (15.4%)
Previous treatment	
*RFA* *Liver surgery* *Electroporation* *Chemotherapy* *Radiotherapy* *TACE* *ICI* *Targeted therapy*	4 (17.4%) 9 (39.1%) 2 (8.7%) 20 (87.0%) 3 (13.0%) 1 (4.3%) 1 (4.3%) 6 (26.1%)
Localization	
*Left liver* *Segment 4* *Segment 1* *Right liver*	5 (16.1%) 9 (29.0%) 2 (6.5%) 15 (48.4%)

RFA = Radiofrequency Ablation, TACE = Transarterial Chemoembolization, ICI = Immune Checkpoint Inhibitors.

**Table 2 jcm-12-01183-t002:** Dosimetric data for initial plans.

Characteristics	Number of Lesions (%) or Median Value (Min–Max)
Total Dose (Gy)	
*60* *50* *40* *35* *30*	2 (6.5%) 3 (9.7%) 1 (3.2%) 17 (54.8%) 8 (25.8%)
Total treatment duration (days)	5.0 (5.0–29.0)
Fraction dose (Gy)	10 (8–12)
Median PTV (cm^3^)	35.6 (9.9–343.2)
Median liver volume (cm^3^)	1372.3 (676.9–2158.5)
Fraction duration (minutes)	82.6 (52–133)
PTV	
*V95% (%)* *V100% (%)* *D95% (Gy)* *D1cc (Gy)*	95.9 (71.6–98.9) 50.0 (49.2–90.8) 47.5 (26.9–58.0) 52.0 (41.2–63.1))
Kidney	
*V18 Gy (cm^3^)*	0.0 (0.0–0.4)
Spinal Cord	
*Dmax (Gy)*	6.9 (0.7–20.3)
Stomach	
*Dmax (Gy)*	17.1 (0.4–31.5)
Duodenum	
*Dmax (Gy)*	11.4 (0.0–27.4)
Small intestine	
*Dmax (Gy)*	3.5 (0.0–29.0)
Large intestine	
*Dmax (Gy)*	12.5 (0.3–31.8)
Esophagus	
*Dmax (Gy)*	6.5 (0.0–23.9)
Heart	
*Dmax (Gy)*	23.0 (0.7–38.0)
Liver	
*Mean dose* *V15 Gy (cm^3^)*	8.5 (3.7–19.1) 342.2 (63.5–774.2)

**Table 3 jcm-12-01183-t003:** Acute and late clinical events following liver MRgRT.

CTCAE v5.0	Acute Toxicity (0–90 Days) (26 Patients)	Late Toxicity (90 Days–1 Year) (22 Patients)
*Abdominal pain*		
g0 g1 g2	22 (84.8%) 2 (7.6%) 2 (7.6%)	14 (63.6%) 3 (13.6%) 5 (22.8%)
*Nausea/Vomiting*		
g0 g1 g2	17 (69.3%) 7 (26.9%) 1 (3.8%)	19 (86.3%) 2 (9.1%) 1 (4.6%)
*Gastro-Duodenal ulcer*		
g0 g1	26 (100%) 0 (0%)	21 (95.4%) 1 (4.6%)
*Parietal pain*		
g0 g1 g2	22 (84.8%) 3 (11.4%) 1 (3.8%)	22 (100%) 0 (0%) 0 (0%)
*Diarrhea*		
g0 g1 g2	24 (92.3%) 2 (7.7%) 0 (0%)	18 (81.8%) 2 (9.1%) 2 (9.1%)
*Ascites*		
g0	26 (100%)	22 (100%)
*Biological hepatic cytolysis*		
g0 g1 g2	25 (96.2%) 1 (3.2%) 0 (0%)	16 (72.7%) 6 (27.3%) 0 (0%)
*Other gastrointestinal and hepatobiliary events*		
Angiocholitis (g3) Angiocholitis (g4) Bile duct stenosis (g4)	1 (3.8%) 1 (3.8%) 0 (0%)	2 (9.1%) 0 (0%) 1 (4.6%)

## Data Availability

Data can be made available upon request with appropriate approval by the investigator team and research administrative offices. Data sharing will be subject to appropriate data transfer agreements. Data requests can be made to the corresponding author after manuscript publication.

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
