# Peer review of "Stereotactic MR-Guided Radiotherapy for Liver Metastases: First Results of the Montpellier Prospective Registry Study"

_jcm, 2023, doi:10.3390/jcm12031183_

Round 1
Reviewer 1 Report
The abstracts and first page are a mess. Fonts in different sizes. Both in the abstract and in the introduction, it has an acronym that is OAR. The authors did not spell out what this means. Correct this.
On material and methods, the authors. included the passage between lines 78 and 80:
“Inclusion criteria were: synchronous or metachronous oligo-metastatic liver metastases, oligo-progressive liver metastases, all from various primary cancers.”
It would be interesting to better explain the reasons for using this approach and what are its advantages.
At the end of the results, the authors quote Appendix 5, which I could not find in the text or in the attached files. In addition, only Appendix 5 is cited, and the Appendixes with previous numbers do not exist. The reader's feeling is that the text is a mess. It looks like it was copied and pasted with lots of different formatting. The text also seems to lack a better connection between one paragraph and another.
Furthermore, introduction, material and methods, discussion and conclusion can and should be improved.
Reviewer 2 Report
This manuscript presents the first clinical results from a prospective registry of Stereotactic MR-Guided Radiotherapy (MRgRT) for liver metastases. Primary endpoints were acute and late toxicities. Secondary endpoints were survival outcomes. The authors included 26 consecutive patients who were treated for 31 liver metastases. They concluded that overall the treatment was well-tolerated and achieved high LC rate.
I found the study well written and well conducted, despite its low sample size. I have no major concerns to be addressed. My only doubt is regarding the use of the term "prospective" for the methodology of this study throughout the whole text, but in Discussion the authors mention "In addition to the retrospective-ambispectival, monocentric and non-randomized nature of the study". Is this retrospective or prospective? Could the authors make this clearer?
Round 2
Reviewer 1 Report
The authors show a manuscript with very interesting data, which I believe will also be of interest to the audience. In the beginning, they presented a first version that was a little to be desired. However, following the reviewers' recommendations, the authors significantly improved the manuscript. In my opinion, the manuscript can be approved for publication as an article.